# Seeking to Develop Global SYK-Ness

**Dmitri V. Khveshchenko**

Department of Physics and Astronomy, University of North Carolina, Chapel Hill, NC 27599, USA;
khvesh@physics.unc.edu, Tel.: +1-919-962-7213

**Abstract:** Inspired by the recent interest in the Sachdev–Ye–Kitaev (SYK) model, we study a class of multi-flavored one- and two-band fermion systems with no bare dispersion. In contrast to the previous work on the SYK model that would routinely assume spatial locality, thus unequivocally arriving at the so-called 'locally-critical' scenario, we seek to attain a spatially-dispersing 'globally-SYK' behavior. To that end, a variety of the Lorentz-(non)invariant space-and/or-time dependent algebraically decaying interaction functions is considered and some of the thermodynamic and transport properties of such systems are discussed.

**Keywords:** strongly correlated fermions; SYK model; holography

## 1. Introduction

The recent rise of the asymptotically solvable $0 + 1$-dimensional SYK model [1–4] possessing a genuine (albeit, still debated over its fine details) holographic dual has also rekindled the long-standing interest in analytically solvable examples of non-Fermi liquid (NFL) behavior. To that end, References [5–23] have extensively explored the idea of combining multiple quantum dot-like copies of the SYK model and/or hybridizing them with some itinerant fermions. However, all those works operated under the common assumption of a spatially local nature of the SYK propagator which limitation resulted in a number of the NFL scenarios exhibiting '(ultra-)local' criticality characterized by the lack of any spatial dispersion.

In view of its formal affinity to the well established dynamical mean-field theory and some resemblance to the observed properties of such families of strongly correlated compounds as the cuprates, pnictides, and heavy fermion materials the (ultra-)local regime was claimed to be physically relevant and possibly providing a new evidence in support of the intriguing, yet still largely speculative, 'bottom-up' holographic phenomenology [24–26].

However, it would seem that a firm justification of the general holographic approach as a viable phenomenological scheme should require one to venture off the beaten path by attempting to extend it to the multitude of more general, including spatially dispersing, NFLs. A host of such states were generated in the previous holographic studies involving the so-called Lifshitz and hyperscaling-violating background geometries and numerous opportunistic proposals for utilizing them in the studies of materials were put forward (although the seemingly endless flurry of those look-alike exercises in classical Einstein–Maxwell-dilaton relativity has been finally withering out, as of lately).

In that regard, a further investigation into the various generalizations of the SYK model and its non-random cousins [27–30] could pave the way for gaining insight into such largely uncharted territory as the various 'super-strongly coupled' systems with a nearly—or even completely—flat dispersion. Their dynamics is then governed solely by the (in general) space- and/or time-dependent $q$-fermion couplings which can, among other things, endow the bare flat-band fermions with a non-trivial dispersion $\epsilon(p)$.

The pertinent interaction functions—-which play the role of disorder correlations in the (generalized) SYK models [2–23]—could then be distinguished on the basis of such important properties as their short- vs long-ranged nature, combined vs. independent dependence on space-time separation, etc. Depending on such important details, the systems in question might demonstrate an entire variety of novel (non)critical regimes.

As the SYK-related examples show, the key requirement that enables asymptotically exact solutions of such systems—as well as their non-random counterparts [27–30]—is a large number $N$ of the different species (see, however, Equation (6) below for a further clarification).

## 2. Results

### 2.1. Model

In what follows, we consider the $d + 1$-dimensional non-random action

$$S = \int_\tau \sum_i^L \sum_\alpha^N \chi_i^{\alpha\dagger} \partial_\tau \chi_i^\alpha - \frac{i^{q/2}}{N^{q-1}} \int_{\tau_k} \sum_{i_l}^L \sum_{\alpha_m,\beta_n}^N F_{i_1...i_q,j_1...j_q}^{\alpha_1...\alpha_q\beta_1...\beta_q} \prod_{k=1}^{[q/2]} \chi_{i_k}^{\alpha_k\dagger} \prod_{l=[q/2]+1}^q \chi_{i_l}^{\alpha_l} \prod_{m=1}^{[q/2]} \chi_{j_m}^{\beta_m\dagger} \prod_{n=[q/2]+1}^q \chi_{j_n}^{\beta_n}, \quad (1)$$

where the Greek indexes stand for the $N$-valued colors, while the Latin ones run over the $L$ sites of a $d$-dimensional cubic lattice. In addition, for the sake of simplicity and in order to focus on the most interesting regime, we choose the chemical potential of the complex fermions $\chi$ to be $\mu = 0$, thus enforcing particle-hole symmetry.

To introduce spatial (and/or additional temporal) dispersion into the problem, while keeping it tractable, we consider a broad family of interaction functions

$$F_{i_1...i_q j_1,...j_q}^{\alpha_1...\alpha_q\beta_1...\beta_q}(\tau) = F_{\mathbf{x}_{ij}}(\tau) \prod_a \delta^{\alpha_a\beta_a} \delta_{i_a i} \delta_{j_a j}, \quad (2)$$

which depend algebraically on the separation in space and/or time between the common locations of simultaneously (dis)appearing $q$-fermion complexes, which is

$$F_{\mathbf{x}}(\tau) = \frac{F}{|\tau|^{2\alpha} |\mathbf{x}|^{2\beta}}. \quad (3)$$

Since in a lattice model the distance $\mathbf{x}_{ij}$ takes discrete values, Equation (3) needs to be carefully defined at its shortest values.

In the generalization of the original SYK model proposed in Reference [31], action (1) results from taking the Gaussian average over the (completely antisymmetric under the simultaneous permutations $\alpha_a \leftrightarrow \alpha_b$ and $i_a \leftrightarrow i_b$) random amplitudes that entangle the groups of $q$ fermions. By choosing $F_{\mathbf{x}_{ij}}(\tau) \sim \delta_{ij}$, one obtains $L$ decoupled copies of the original $N$-colored SYK model, while by allowing for non-zero nearest-neighbor terms $F_{\mathbf{x}_{ij}}(\tau) \sim \delta_{i,j+\mathbf{e}}$ one can describe the various SYK lattice (chain) models of References [5–23].

In contrast, an instantaneous and/or contact interaction corresponds to choosing $\alpha = 1/2$ and/or $\beta = d/2$, which power count is similar to that of a $\delta$-function in the time- and/or space-domain. It should be pointed out, though, that a uniform and spacetime-independent correlation function $F_{\mathbf{x}_{ij}}(\tau) = const$ ($\alpha = \beta = 0$) does not reproduce the original (single-site) SYK model of the total of $NL$ fermions where the entangling correlations would be equally strong among any $2q$ orbitals chosen arbitrarily from any of the $L$ sites and $N$ colors alike.

Notably, in the limit of $\beta \to 0$, the theory (1) becomes symmetric under the permutations among the $L$ sites, so the very notion of a spatial distance, alongside an underlying lattice structure, becomes ill-defined. Nevertheless, as long as the correlation amplitude (3) remains distance-dependent, the fermion correlations acquire an emergent non-trivial dispersion, as demonstrated below.

## 2.2. Schwinger–Dyson Equations and Their Solutions

Analogously with the previous analyses of the SYK-type models [2–31], the partition function of the theory (1) can be written as the path integral over a pair of bosonic fields $G$ and $\Sigma$ whose expectation values yield the fermion propagator and self-energy, respectively:

$$Z = \int DG(\tau, \mathbf{x}) D\Sigma(\tau, \mathbf{x}) Det(\delta(\mathbf{x})\partial_\tau + \Sigma(\tau, \mathbf{x})), \exp\left(\frac{N}{2} \int_{\tau, \mathbf{x}} \left(\frac{1}{q} F_\mathbf{x}(\tau) G^q(\tau, \mathbf{x}) - G(\tau, \mathbf{x})\Sigma(\tau, \mathbf{x})\right)\right), \quad (4)$$

where the determinant results from integrating out the fermions and, with a focus on the IR regime, the discrete sum over the lattice sites can be replaced with the integral over the spatial coordinate $\mathbf{x}$.

The saddle points in theory (4) corresponds to the various solutions of the Schwinger–Dyson (SD) equation

$$\partial_{\tau_1} G(\tau_{12}, \mathbf{x}_{12}) + \int_{\tau_3, \mathbf{x}_3} \Sigma(\tau_{13}, \mathbf{x}_{13}) G(\tau_{32}, \mathbf{x}_{32}) = \delta(\mathbf{x}_{12})\delta(\tau_{12}), \quad (5)$$

where the self-energy is given, to leading order in $1/N$, by the sum of the so-called 'watermelon' diagrams while ignoring any vertex corrections

$$\Sigma(\tau, \mathbf{x}) = F_\mathbf{x}(\tau) G^{q-1}(\tau, \mathbf{x}). \quad (6)$$

It must be noted, though, that a solid justification of the approximation behind Equation (6) may require some adjustments to the action (1), thus effectively making it conform to one of the non-random '(non)colored tensor' models of References [27–30]. In practice, it can be achieved by decorating each of the $L$ sites of the underlying lattice with a properly designed unit cell composed of $N$ sites which are occupied by the $q$ fermions at split locations (see Reference [23] for an explicit example of such construction). However, this technical complication appears to have no bearing on the robust algebraically decaying amplitudes that we are going to study.

The Fourier transform of Equations (5) and (6) then reads

$$G^{-1}(\epsilon, \mathbf{p}) = i\epsilon + \prod_{i=1}^{q-1} \int_{\omega_i, \mathbf{k}_i} G(\omega_i, \mathbf{k}_i) F_{\mathbf{p}+\sum_i \mathbf{k}_i}\left(\epsilon + \sum_i \omega_i\right). \quad (7)$$

A relative importance of the interaction-induced self-energy can be ascertained by the standard arguments. Under a scaling of the temporal, $\omega \to s\omega$, and spatial, $k \to s^{1/z}k$, dimensions, where $z$ is the dynamical critical exponent, one finds that the self-energy dominates over the kinetic term or, at least, remains marginally relevant in both the UV and IR regimes, provided that the condition

$$\frac{d(q-2) + 2\beta}{z} + 2\alpha - 2 \leq 0 \quad (8)$$

is met [31].

We leave a systematic investigation into all the viable solutions of Equation (7) to future work. Such solutions should ultimately be selected on the basis of their (minimal) energies—for a reliable evaluation of which a proper ansatz first needs to be chosen. Such choice is likely to depend on the details of the action (1) and, therefore, may not be universally applicable.

As was first argued in the case of the random SYK-type models of Reference [31], the customary ultra-local solution $G(\tau, \mathbf{x}) = 0$ for $\mathbf{x} \neq \mathbf{0}$ (hence, $G(\epsilon, \mathbf{p}) = G(\epsilon)$) would generally be favored by the Hartree-type terms in the overall fermion energy, whereas the Fock-type ones tend to support non-local ones. Moreover, while being finite when evaluated on the ultra-local solution in the case of short-ranged couplings, the Hartree terms develop IR divergences, once the fermion interactions become sufficiently long-ranged.

For instance, the lattice sum $\sum_\mathbf{x} F_\mathbf{x}(\tau)$ appearing in the Hartree terms with $F_\mathbf{x}(\tau)$ given by Equation (3) diverges for all $\beta \leq d/2$ (in contrast, a spurious UV divergence for $\beta > d/2$ is absent as

long as the separately defined amplitude $F_0(\tau)$ remains finite). This observation alone suggests that, at least for $\beta \leq d/2$, the ultra-local solution becomes unstable, as compared to a non-local one.

Therefore, in the vicinity of an emergent fermion dispersion ('on-shell'), we seek the solution of Equation (7) in the general form

$$G(\epsilon, \mathbf{p}) = \frac{A(\mathbf{p})}{(i\epsilon - Bp^z)^\eta},\tag{9}$$

for which ansatz includes all the important ingredients: effective dispersion relation characterized by the critical exponent $z$ and prefactor $B$, anomalous exponent $\eta$ controlling the 'on-shell' singularity, and the 'wave-function renormalization' $A(\mathbf{p})$. Away from the 'on-shell' regime (be it at an extended Fermi surface or near an isolated nodal Dirac point), the ansatz (9) is no longer applicable, so its use can only be justified if the integrals in Equation (7) are dominated by the 'on-shell' contributions—which indeed appear to be the case.

Analyzing Equation (7) in the 'on-shell' regime $|\epsilon - Bp^z| \ll \omega, Bp^z$ and equating the powers of $\epsilon$ and $\mathbf{p}$, alongside dimensionfull prefactors, on both sides, one finds the anomalous dimension $\eta$

$$\eta = 1 - \frac{1 - 2\alpha}{q},\tag{10}$$

together with the dynamical critical exponent

$$z = \frac{d(q - 2) + 2\beta}{2 - 2\alpha}\tag{11}$$

as well as the residue

$$A(\mathbf{p}) \propto (F\mathbf{p}^{d(q-2)+2\beta})^{\frac{(2\alpha-1)}{q(2-2\alpha)}}\tag{12}$$

and the coefficient $B \sim F^{\frac{1}{2-2\alpha}}$ in the emergent fermion dispersion. A non-vanishing value of the latter implies that the propagator $G(\epsilon, \mathbf{p})$ acquires a non-trivial momentum dependence, thus signaling that its real-space Fourier transform is no longer ultra-local.

Alternatively, this can be viewed as a spontaneous breaking of the local $Z_2$ symmetry $\chi_i^\alpha \to -\chi_i^\alpha$ of the action given by Equations (1) and (2) which, if preserved, would have prohibited any spatial non-locality, thus enforcing the condition of ultra-locality, $< \chi_i^\dagger \chi_j > = 0$ for $i \neq j$.

When contemplating the general possibility of such symmetry breaking, it is worth noting that it is, in fact, specific to the two-point interaction function given by Equation (2), whereas for a generic $F_{i_1...i_q j_1,...j_q}^{\alpha_1...\alpha_q \beta_1...\beta_q}$ Equation (1) would not possess this symmetry in the first place.

As regards the appicability of the above solution, the criterion (8) appears to be satisied as equality, thus signifying a marginally relevant nature of the $2q$-fermion interaction given by Equations (2) and (3). Among other things, this makes it difficult to find a numerical prefactor in Equation (12), as both, the self-energy and bare kinetic, terms in Equation (7) turn out to be important.

In what follows, we restrict our analysis to the parameter values $0 \leq \alpha \leq 1/2$ for which $\eta < 1$ and the propagator (9) readily conforms to the anticipated 'no quasiparticle' regime. Moreover, the above restriction also guarantees that the physical condition $z > 0$ will be fulfilled for any $d \geq 0$ and $q > 1$ as long as $\beta \geq 0$.

In particular, for $q = 2$ and in the limit $\alpha = \beta \to 0$, the theory (1) is equivalent to the disorder-averaged action for a non-interacting single-particle state of a fixed energy. Then, Equation (7) becomes merely algebraic and features an exact momentum-independent solution

$$G(\epsilon, \mathbf{p}) = \frac{2}{i\epsilon + \sqrt{(i\epsilon)^2 - F}},\tag{13}$$

which, upon being expanded in the on-shell regime ($|i\epsilon - F^{1/2}\mathbf{p}^0| \ll |i\epsilon|$), manifests the exponents $\eta = 1/2$ and $z = 0$, as well as the coefficients $A \sim F^{-1/4}\mathbf{p}^0$ and $B \sim F^{1/2}$, in full agreement with

Equations (10)–(12). Notably, though, in the special case of $q = 1$, action (1) becomes Gaussian and the exact fermion propagator can be immediately obtained as the inverse of the quadratic kernel

$$G(\epsilon, \mathbf{p}) = \frac{(i\epsilon)^{1-2\alpha}}{(i\epsilon)^{2-2\alpha} + F\mathbf{p}^{2\beta-d}}. \tag{14}$$

This expression exhibits the dynamical index $z = (2\beta - d)/(2 - 2\alpha)$ which is again consistent with (11), although, under the previously imposed restriction $0 \le \alpha \le 1/2$, the condition $z > 0$ now requires a convergence of the spatial Fourier transform of Equation (3), i.e., $\beta \ge d/2$.

Thermodynamics of the system described by Equation (1) can be studied with the use of the Luttinger–Ward functional which yields the free energy

$$\frac{\mathcal{F}}{N} = \int_{\tau, \mathbf{x}} (\ln G^{-1} + G\Sigma - \frac{1}{q}FG^q) \sim T^{1+d/z}. \tag{15}$$

For $d = 0$ and/or $z = \infty$, it manifests such a salient feature of the SYK physics as finite zero-temperature entropy, $S(T) = -\frac{\partial F}{\partial T} \sim T^{d/z}$, whereas for $d > 0$ and $z < \infty$ the result complies with the 3rd law of thermodynamics.

Transport coefficients such as longitudinal optical conductivity can also be evaluated in the 'no vertex correction' approximation (cf. References [32–34])

$$\sigma(\omega) = \frac{z^2}{d\omega} \int_{\epsilon, \mathbf{p}} \frac{\epsilon(\epsilon + \omega)}{\mathbf{p}^2} G(\epsilon + \omega, \mathbf{p}) G(\epsilon, \mathbf{p}) \sim \omega^{2-2\eta+(d-2)/z}. \tag{16}$$

For $\alpha = \beta = 0$ and $z \to \infty$, the exponent in Equation (16) becomes $4/q$ which contains the extra factor of $\omega^2$ as compared to the result of Reference [34] where the fermions were endowed (despite their purportedly localized nature) with a bare dispersion characterized by a finite velocity $v_F$.

However, Equation (16) does pass one important check: for $d = 2$, $T = 0$, and in the presence of well-defined quasiparticles ($\eta = 1$) the conductivity $\sigma(\omega)$ becomes a dimensionless constant as in this case there are no relevant energy scale that could be combined with the frequency into one dimensionless ratio. In turn, the vanishing in the D.C. limit ('soft gap') behavior of Equation (16) for $d \ge 2$ and $\eta < 1$ would be consistent with the highly correlated nature of the system in question.

Another potentially interesting class of models is represented by the manifestly Lorentz-invariant interaction functions

$$F_{\mathbf{x}}(\tau) = \frac{F}{|\tau^2 - \mathbf{x}^2|^\gamma}. \tag{17}$$

In this case, the fermion propagator inherits the same symmetry through Equation (7), thus forcing the dynamical exponent $z = 1$ upon the solution

$$G(\epsilon, \mathbf{p}) = \frac{A}{((i\epsilon)^2 - \mathbf{p}^2)^{\eta/2}}, \tag{18}$$

which is manifestly spatially non-local. This time around, equating the powers of $\epsilon$ and $\mathbf{p}$ on both sides of the Equation (10) yields

$$\eta = d + 1 - 2\frac{d+1-\gamma}{q}, \tag{19}$$

while the strong-coupling regime can now be attained under the condition

$$d(q-2) + 2\gamma - 2 \le 0, \tag{20}$$

which turns out to be rather restrictive. In particular, a contact instantaneous coupling of the Thirring or Gross–Neveu variety which scales similarly to the power-law with $\gamma = (d+1)/2$ limits the applicability of Equation (19) to $d = 1, q = 2$ where it appears to be, at most, marginal

(cf. References [35–37]). Nonetheless, for $q = 2$ and $\gamma \leq 1$, the 'no quasiparticle' condition can hold for all $d$. In addition, for the maximally (space-time) long-ranged 3-particle couplings ($q = 3$) with $\gamma = 0$, the upper critical dimension that fulfills the criterion (20) can be as high as $d = 2$.

## 2.3. Hybrid Models

In addition of interest are the two-band models where the SYK fermions ($\chi$) are coupled to some 'itinerant' ($\psi$) ones, the latter possessing a dispersion $\xi_{\mathbf{k}} \approx v_F(k - k_F)$

$$S = \int_\tau \sum_{i,j}^L \sum_\alpha^N [\delta_{ij}\chi_i^{\alpha\dagger}\partial_\tau\chi_j^\alpha + \psi_i^{\alpha\dagger}(\delta_{ij}\partial_\tau - \xi_{ij})\psi_j^\alpha]$$

$$-\frac{i^{q/2}}{N^{(q-1)}} \int_{\tau_k} \sum_{i_l}^L \sum_{\alpha_m,\beta_n}^N F_{i_1\ldots i_q,j_1\ldots j_q}^{\alpha_1\ldots\alpha_q\beta_1\ldots\beta_q} \prod_{k=1}^{[q/2]} \chi_{i_k}^{\alpha_k\dagger} \prod_{l=[q/2]+1}^q \chi_{i_l}^{\alpha_l} \prod_{m=1}^{[q/2]} \chi_{j_m}^{\beta_m\dagger} \prod_{n=[q/2]+1}^q \chi_{j_n}^{\beta_n} \qquad (21)$$

$$-\frac{i^{p/2}}{N^{(p-1)}} \int_{\tau_k} \sum_{i_l}^L \sum_{\alpha_m,\beta_n}^N H_{i_1\ldots i_p,j_1\ldots j_p}^{\alpha_1\ldots\alpha_p\beta_1\ldots\beta_p} \prod_{k=1}^{[p/2]} \chi_{i_k}^{\alpha_k\dagger} \prod_{l=[p/2]+1}^p \chi_{i_l}^{\alpha_l} \prod_{m=1}^{[p/2]} \psi_{j_m}^{\beta_m\dagger} \prod_{n=[p/2]+1}^p \psi_{j_n}^{\beta_n},$$

where the functions $F$ and $H$ describe, respectively, a $q$-particle self-interaction of the SYK fermions (*chi*) and a $p$-particle coupling between the SYK and itinerant ($\psi$) ones.

The coupled SD equations now read

$$\Sigma_\chi(\tau, \mathbf{x}) = F_{\mathbf{x}}(\tau)G_\chi^{q-1}(\tau, \mathbf{x}) + H_{\mathbf{x}}(\tau)G_\chi^{p/2-1}(\tau, \mathbf{x})G_\psi^{p/2}(\tau, \mathbf{x}) \qquad (22)$$

and

$$\Sigma_\psi(\tau, \mathbf{x}) = H_{\mathbf{x}}(\tau)G_\psi^{p/2-1}(\tau, \mathbf{x})G_\chi^{p/2}(\tau, \mathbf{x}). \qquad (23)$$

In what follows, for simplicity, we choose the function $H_{\mathbf{x}}(\tau)$ to be given by the same Equation (3), albeit with an independent prefactor.

For $p = 2$, the corresponding self-energies generated by the $H$-coupling can be evaluated as

$$\Sigma_{\chi,\psi}^H(\omega, \mathbf{k}) = \int_{\epsilon,\mathbf{p}} G_{\psi,\chi}(\omega + \epsilon, \mathbf{p} + \mathbf{k})H_{\mathbf{p}}(\epsilon). \qquad (24)$$

A self-consistent solution of the coupled Equation (24) shows that the contribution $\Sigma_\chi^H \sim \omega^\eta$ towards the total self-energy of the $\chi$-fermions is of the same functional form (9) as that of ($\Sigma_\chi^F$) due to the self-interaction between them. Concomitantly, the spectrum of the $\psi$-fermions gets strongly affected by their coupling to the $\chi$-ones

$$\Sigma_\psi^H(\epsilon) \sim \epsilon^{2\alpha-\eta+2\beta/z}. \qquad (25)$$

In the case of $F_{\mathbf{k}}(\omega), H_{\mathbf{k}}(\omega) \propto \delta(\omega)const(\mathbf{k})$ ($\alpha = 0, \beta = d/2$) and by putting $z = \infty$ as in the SYK model, one can readily reproduce the results of Reference [32]: $\Sigma_\chi^H \sim \omega^{1-2/q}$ and $\Sigma_\psi^H \sim \omega^{2/q-1}$ which manifest a markedly incoherent behavior of the $\psi$-fermions.

In the other practically important case of $p = 4$, one obtains

$$\Sigma_{\chi,\psi}^H(\omega, \mathbf{k}) = \int_{\Omega,\epsilon,\mathbf{q},\mathbf{p}} G_{\chi,\psi}(\Omega + \epsilon + \omega, \mathbf{p} + \mathbf{k} + \mathbf{q})\Pi_{\psi,\chi}(\Omega, \mathbf{q})H_{\mathbf{p}}(\epsilon), \qquad (26)$$

where the polarization operators

$$\Pi_{\psi,\chi}(\omega, \mathbf{k}) = \int_{\epsilon,\mathbf{p}} G_{\psi,\chi}(\epsilon + \omega, \mathbf{p} + \mathbf{k})G_{\psi,\chi}(\epsilon, \mathbf{p}) \qquad (27)$$

have to be evaluated self-consistently with the use of the exact propagators.

Performing this procedure under the assumption that the correction (24) does not alter the functional form of the overall self-energy $\Sigma_\chi^H + \Sigma_\chi^F$ which is still given by Equation (9), one obtains

$$\Sigma_\psi^H(\epsilon) \sim \epsilon^{1-\eta+\alpha+(d+\beta)/z}. \tag{28}$$

Conversely, with the use of (28), one can confirm that the dynamics of the $\chi$-fermions is governed by both their self-interaction and coupling to the $\psi$-ones, thus justifying the above assumption.

In the case of $q = 4$, $F_{\mathbf{k}}(\omega), H_{\mathbf{k}}(\omega) \propto \delta(\omega)const(\mathbf{k})$ ($\alpha = 0, \beta = d/2$) and $z = \infty$, the above predictions can be contrasted against the results of Reference [29]: $\Sigma_\chi \sim \omega$ and $\Sigma_\psi \sim \omega \ln \omega$. In turn, the results of Reference [34], $\Sigma_\chi \sim \omega^{1+2/q}$ and $\Sigma_\psi \sim \omega^{4/q} \ln \omega$, pertain to the case of $q \geq 4$, $F_{\mathbf{k}}(\omega), H_{\mathbf{k}}(\omega) \propto \delta(\omega)\delta(\mathbf{k})$ ($\alpha = \beta = 0$), and $z = \infty$. It should be noted, though, that the above estimates originate from using the free-fermion polarization operator $\Pi_\psi(\omega, \mathbf{k}) \propto 1 + \omega/\sqrt{\mathbf{k}^2 + \omega^2}$ instead of the proper one at all (rather than only small) momenta in Equation (26). Such approximation is prone to lacking self-consistency which would be pertinent to the asymptotic strong-coupling regime.

In fact, had one chosen to use the bare polarization function $\Pi_\psi$ in (24), while maintaining $z = \infty$, it would have resulted in the expressions $\Sigma_\chi^H \sim \omega^{1-\eta}$ and $\Sigma_\psi^H \sim \omega^{2-2\eta}$ which get augmented by the factors of $\ln \omega$ if the corresponding integrals are logarithmic, just as in the aforementioned cases.

Thermodynamics of the two-band model is described by the Luttinger–Ward functional that includes Equation (15) written in terms of $G_\chi$, alongside the additional term

$$\frac{\Delta \mathcal{F}}{N} = \int_{\omega,\mathbf{k}} (\ln G_\psi^{-1} + G_\psi \Sigma_\psi - \frac{1}{p} H G_\psi^{p/2} G_\chi^{p/2}). \tag{29}$$

Computing (29) with the use of the exact propagators that utilize Equations (9) and (28), one arrives at the same behavior (15), which, once again, signifies the lack of a competing energy scale in the IR strong-coupling regime $T \ll F^{1/2}$.

As to the conductivity of the $\psi$-fermions, one now obtains

$$\sigma_\psi(\omega) = \frac{1}{\omega d} \int_{\epsilon,\mathbf{p}} v_{\mathbf{p}}^2 G_\psi(\epsilon + \omega, \mathbf{p}) G_\psi(\epsilon, \mathbf{p}) \sim \omega^{2\eta - 2 - 2\alpha - (d+2\beta)/z}, \tag{30}$$

which diverges at $\omega \to 0$ for $\eta < 1$, consistent with the general expectation of a system with a non-flat dispersion but no source of momentum relaxation. In addition, should the customary substitution $\omega \to T$ prove to be justifiable (which might indeed be true in the 'no momentum drag' regime), the resistivity would then show a rising ('metallic') behavior with increasing temperature.

In particular, the above results imply that, for $\alpha = \beta = 0$ and $z = \infty$, the exponent in Equation (30) equals $-4/q$, thus suggesting an 'accidentally linear' temperature dependence of resistivity for $q = 4$ (cf. References [34]). However, the general possibility of a host of other values of this exponent should be viewed as a caution against invoking the generalized SYK models to explain the ostensibly 'universal' linear resistivity observed in a variety of strongly correlated systems [32,33].

Once a (non-universal) mechanism of momentum relaxation is specified and added to the Hamiltonian, then frequency- and temperature-dependent conductivity as well as other transport coefficients can be evaluated and tested for a compliance (or a lack thereof) with such hallmarks of the Fermi liquid regime as the Wiedemann–Franz, Mott, and other standard relations. Such a systematic analysis would help one to ascertain a potential viability of the 'globally SYK' models for describing the phenomenology of certain strongly correlated materials. We leave these tasks to future work.

In the two-band models, the potentially competing long-ranged $q$- and $p$- body interactions might drive various transitions from the parent—diffusive and highly chaotic ('incoherent metallic')—state to either a ('heavy') Fermi liquid, a (many-body) localized insulator, or some ordered states, all with a varying temperature and/or rates of decay and strengths of the couplings. Therefore, it would be interesting to carry out a systematic analyses of the potential instabilities of action (21).

In addition, as far as the current global quest into holography is concerned, it would be an interesting challenge to come up with a plausible conjecture for a holographic dual of such models (see Reference [38] for a preliminary attempt in that direction). For one thing, the viable background geometry is expected to be rid of the ubiquitous near-extremal black hole with its universal $AdS_2 \times R^d$ metric as in its presence the boundary state would likely remain maximally chaotic. Indeed, the multi-dimensional and long-ranged generalization of the random SYK model studied in Reference [31] was found to be less chaotic than the original one and also sub-diffusive, thus diminishing the chances of its having an $AdS_2$ (or, for that matter, any) type of a holographic dual.

## 3. Conclusions

To summarize, in the present communication, an exploratory attempt was made to investigate the properties of 'super-strongly coupled' fermion systems with no bare dispersion and long-ranged interactions. Certain Lorentz (non-)invariant solutions of the corresponding SD equations were proposed and some basic properties of such systems discussed, including the continuously varying exponents in the power-law behavior of spectral function, entropy, and optical conductivity.

This study continues the investigation of Reference [31] into the question of whether or not the 'holography-friendly' properties of the original SYK model can survive such an important generalization as a non-trivial time/space dependence. If that were the case, one could argue that the SYK model describes generic properties of a whole universality class of systems, thereby providing much needed support for a possible applicability of the tantalizing 'bottom-up' holographic approach to a broad variety of quantum systems.

Instead, the pristine SYK behavior appears to be fragile and hardly extendable beyond the original SYK model and, possibly, its minimal (irrelevant-operator-driven) modifications, thus leaving the status of the overreaching holographic conjecture as undetermined as it was before the rise of the SYK model.

**Funding:** This research received no external funding.

**Conflicts of Interest:** The author declares no conflict of interest.

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
