# Peer review of "Seeking to Develop Global SYK-Ness"

_condensedmatter, doi:10.3390/condmat3040040_

Round 1
Reviewer 1 Report
Spatial locality is absolutely a crucial ingredient in both condensed matter systems and quantum field theories. It is not very clear what are the physical motivations of the generalized SYK models provided in the current paper.
The definition of the model in Eq. (1) is not clear where are the daggers. For example for i2 and j2, which one has a dagger on them?
After Eq. 12, the author claimed that somehow the system spontaneously breaks fermion parity symmetry which is extremely wired. What is the signature of fermion parity symmetry breaking? Why this system tends to break fermion parity?
It is not at all clear why the scaling form like Eq. (9) would work for solving the saddle point equation. What is the physical significance of the on-shell regime? Why on-shell regime should have a conformal solution?
Can the result of optical conductivity be applied to any physical systems?
Author Response
Author's reply to the comments by Referee I:
Q1."Spatial locality is absolutely a crucial ingredient in both condensed matter systems and quantum field theories. It is not very clear what are the physical motivations of the generalized
SYK models provided in the current paper."
A1: We surmise that the referee might have confused the locality of an abstract field theory (as a general possibility to formulate it in terms of local variables) with the locality of its correlations (as would be manifested by its various correlation functions), the latter one being of essence to the present manuscript. If, however, the referee meant precisely the latter then the answer to his/her question should be as obvious as, e.g., the need to consider (spatially non-local) Coulomb interactions when analyzing any realistic condensed matter system.As to the motivation for introducing non-local couplings between the different sites of a multi-dimensional network built out of many (non-random) systems with the SYK-like dynamics, one immediate answer is that it extends the earlier ideas of Refs.[24-26] to a much greater variety of asymptotically solvable models. Presumably, the undeniable scientific value of any such extension would not need to be spelled out to the readers of a physics journal. Another (arguably, even more fundamental) reason derives from the main goal of the earlier study of Ref.[27] which set out to understand as to whether or not the 'holography-friendly' properties of the original SYK model (where the 2q-fermion 'all-to-all' couplings are strictly independent of any distances and/or time intervals) can survive under a similar generalization - that is, non-trivial time/space dependence. In that case the SYK model would describe generic properties of a whole universality class of systems, thereby providing an important argument towards a (much needed) justification of the (otherwise, largely speculative) holographic approach to the quantum many-body systems.Lastly, the specific choice of algebraically-decaying interaction functions was made because it enables explicit analytic solutions.
Q2."The definition of the model in Eq. (1) is not clear where are the daggers. For example for i2 and j2, which one has a dagger on them?"
A2: Eq.(1) was re-written in order to eliminate any future confusion.
Q3."After Eq. 12, the author claimed that somehow the system spontaneously breaks fermion parity symmetry which is extremely wired. What is the signature of fermion parity symmetry breaking? Why this system tends to break fermion parity?"
A3: Assuming that the referee (who, according to the report, finds the English in the manuscript to be at some need of improvement) actually meant "weird" instead of "wired" (which term could have matched the context, too, provided that the IT slang were used), we surmise that she/he might have again been thinkingof fermion parity in its orthodox (i.e., relativistic field-theoretical) sense.Instead, in the (non-relativistic) lattice fermion systems - such as the nearest-neighbor lattice-SYK model - the notion of fermion parity symmetry pertains merely to the changes under the transformation $\chi_i\to -\chi_i$. If such symmetry indeed remained intact it could have been used as the reason for not considering any solutions where the one-fermion function$<\chi^{\dagger}_i\chi_j>$ is non-zero for $i\neq j$.Therefore, a non-vanishing coefficient B in Eq.(9) implies that the propagator $G(\epsilon,{\bf p})$ acquires a non-trivial momentum dependence, thus signaling that its real-space Fourier transform is no longer ultra-local.As to the general possibility of such symmetry breaking it is worth noting that this symmetry is, in fact, specific to the two-point interaction function given by Eq.(2), whereas for a generic $F_{i_1\dots i_qj_1,\dots j_q}^{\alpha_1\dots\alpha_q \beta_1\dots\beta_q}$ Eq.(1) would not possess such symmetry in the first place.The phenomenon of spontaneous symmetry breaking where a solution has lower symmetry than the underlying action is common to the interacting many-body systems. Moreover, even in the presence of an intact local symmetry (e.g., in a gauge theory) a non-invariant amplitude can be used to deduce gauge-invariant information (e.g., the gauge-invariant poles and cuts of a gauge-non-invariant fermion propagator in QED, QCD, etc.)
Q4."It is not at all clear why the scaling form like Eq. (9) would work for solving the saddle point equation. What is the physical significance of the on-shell regime? Why on-shell regime should have a conformal solution?"
A4: The ansatz of Eq.(9) is, in fact, mostgeneral and includes all the four ingredients that would be customarily used in any discussion of the non-Fermi-liquid behavior: renormalized dispersion relation characterized by the critical exponent $z$ and prefactor $B$,anomalous exponent $\eta$ controlling the 'on-shell' singularity, and the 'wave-function renormalization' $A(p)$. As is well known from the elementary textbooks the 'on-shell' behavior of a single-fermion propagator signifies the presence of well-defined quasiparticle excitations - or a lack thereof.Away from the 'on-shell' regime (be it at a continuous Fermi surface or near an isolated nodal Dirac point) the ansatz (9) ceases to be applicable, so its use would only be justified if the integrals in the SD equation were dominated by the 'on-shell' contributions - which indeed appears to be the case.Then, in the absence of any energy scale other than $\epsilon$ and $Bp^z$, the solution of the SD equations is simply a function of these two dynamical scales - although it is not necessarily conformal (the latter would be a stronger property which is not to be confused with the mere scaling with powers of $\epsilon$ and $p$). However, the validity of Eq.(7) itself is provided by the same condition of a large $N$ as that giving rise to the genuine conformal saddle-point solution in the case of the original SYK model.
Q5."Can the result of optical conductivity be applied to any physical systems?"
A5: In the (by now, published) Refs.[28-30] the finite-temperature counterparts of the optical conductivity were indeed used to speculate about the relevance of the SYK-type physics to the data on some popular 'strange metals'.In contrast, in the present manuscript it was clearly stated that in the absence of a specified (non-universal) mechanism of momentum relaxation no general relation would exist between the optical and finite-temperature D.C. conductivities. However, once such a mechanism (umklapp, impurities, phonons) is introduced, the entire temperature- and frequency-dependent conductivity could be evaluated and the results contrasted against experiment. These tasks are beyond the scopes of the exploratory work presented in the manuscript that is meant to be published as brief report.
Reviewer 2 Report
The author considers a non-local generalization of the SYK model referred to as “global SYK” with one--species of multi-colored fermions with q-range interactions, and two-species or “hybrids” with “itinerant” fermions. The non-locality in space-time is assumed either power like or Lorentzian. Unlike the standard SYK models the q-range interaction is not random. The author derives and analyzes the induced fermionic propagator and self energy in the large N limit through standard bosonization. For some range of parameters for the power like interaction, the fermions develop an emergent dispersion relation. The same is observed for the Lorentzian interaction. The class of models addressed in this ms are interesting, and the author points at various limits which relate to the standard SYK limit, as well as the relevance of his class of models to highly correlated fermionic systems. Also the potential relation of this class of models to holography is intriguing. Most of the derivations in this ms are easily reproducible, although some parts need further elaboration in relation to specific physical applications, e.g. heavy fermions, localized insulators as the author says. Overall, I support the publication of this ms in your journal.
Author Response
Author's reply to comments by Referee II:
"The author considers a non-local generalization of the SYK model referred to as “global SYK” with one--species of multi-colored fermions with q-range interactions, and two-species or “hybrids” with “itinerant” fermions. The non-locality in space-time is assumed either power like or Lorentzian. Unlike the standard SYK models the q-range interaction is not random.
The author derives and analyzes the induced fermionic propagator and self energy in the large N limit through standard bosonization. For some range of parameters for the power like interaction, the fermions develop an emergent dispersion relation. The same is observed for the Lorentzian interaction.
The class of models addressed in this ms are interesting, and the author points at various limits which relate to the standard SYK limit, as well as the relevance of his class of models to highly correlated fermionic systems.
Also the potential relation of this class of models to holography is intriguing. Most of the derivations in this ms are easily reproducible, although some parts need further elaboration in relation to specific physical applications, e.g. heavy fermions, localized insulators as the author says. Overall, I support the publication of this ms in your journal."
We thank the referee for supporting publication of the manuscript. As to the concrete physical applications, this exploratory work (in the brief report format) on the long-ranged generalizations of the (non-random) SYK-type models is only meant to demonstrate the existence of a whole new direction in the SYK-related research by questioning such wide-spread assumptions about the SYK physics as its invariable 'ultra-locality'. As such, the manuscript avoids making any overly optimistic claims about and strained references to the actual physical systems as in, e.g., Refs.[28-30]. That would be premature before first resolving the more fundamental universal aspects of the generalized SYK physics.
Round 2
Reviewer 1 Report
I accept all the modifications provided in the new version except a minor complain about the response to the question 3 in my original report.
I finally understand what the author means by the Z2 symmetry, which is NOT the fermion parity symmetry which flips the sign of ALL fermion operators in the hamiltonian. The Z2 symmetry in the current paper is only flipping the fermion operators for a give site i, or can be viewed as fermion parity of the cluster on a given site. The author should be cautious about the way of presenting the symmetry. Because it is very easy to confuse the "local" Z2 symmetry with the global fermion parity. Fermion parity of an isolated system can never be broken while the local fermion parity can.
Can the author provide some physical intuition for why this interaction can induce non-local green's function while the previous models (for example the coupled chain model in Gu et al.) cannot?
Author Response
We appreciate the quick response by Referee I and his/her constructive attitude
that helps to improve on the presentation in the manuscript. In response to his/her new question
We appreciate the quick response by Referee I and his/her constructive attitude
that helps to improve on the presentation in the manuscript. In response to his/her new question we added a clarification to the manuscript,
we added a clarification to the manuscript, following Eq.(8).
The truth of the matter is that the commonly assumed applicability of ultra-local solutions is
yet to be fully justified even in the case of short-ranged correlations,
such as the SYK-lattice models of Refs.[5,11]. An attentive reader of
the previous publications on this topic could have noticed
(and, probably, would have been disturbed by the fact) that the popular
ultra-local ansatz for the singe-fermion propagator [as in the original (space-less) SYK model]
was POSTULATED, rather than proved, in all (repeat: ALL) of those works.
Other than its technical convenience, the only
argument invoked in a couple of publications was the aforementioned
local $Z_2$ symmetry (which, if it indeed remained intact, would have prohibited
any inter-site fermion correlations, thereby enforcing $<\chi^{\dagger}_i\chi_j>=0$ for $i\neq j$).
A systematic investigation into all the viable solutions of the saddle-point equations
in the generalized SYK-type models is yet to be carried out. Such solutions
should ultimately be selected by comparing their energies - for a reliable evaluation
of which a proper ansatz first needs to be chosen.
However, such choice is likely to depend on the details of the action (1) and, therefore,
may not be universally applicable (or, for that matter, allow for a universal
physical explanation).
As was first argued in the case of the random SYK-type models of Ref.[27],
the ultra-local solution would generally be favored by the Hartree-type terms
in the overall fermion energy, whereas the Fock-type ones tend to support the non-local solutions
of the kind proposed in Ref.[27] and the present manuscript.
Moreover, while being finite when evaluated on the
ultra-local solution in the case of short-ranged couplings,
the Hartree terms develop IR divergences, once the fermion interactions
become sufficiently long-ranged.
For instance, the lattice sum $\sum_k F_{ik}(\tau)$
appearing in the Hartree terms with $F_{ik}(\tau)$ given
by Eq.(3) diverges for all $\beta\leq d/2$
(in contrast, a spurious UV divergence for $\beta>d/2$
is absent as long as the separately introduced amplitude $F_{ii}(\tau)$ remains finite).
This observation alone suggests that, at least,
for $\beta\leq d/2$ the ultra-local solution becomes unstable,
as compared to a non-local one.